# Effect of bovine leukemia virus (BLV) infection on bovine mammary epithelial cells RNA-seq transcriptome profile

**Lucia Martinez Cuesta[1], Juan Pedro Liron[2], María Victoria Nieto Farias[1], Guillermina Laura Dolcini[1], Maria Carolina Ceriani[1] ***

**1** Virology, SAMP Department, Centro de Investigación Veterinaria de Tandil (CIVETAN), Consejo Nacional de Investigaciones Científicas y Técnicas (CONICET), Tandil, Buenos Aires, Argentina, **2** Pharmacology, FISFARVET Department, Centro de Investigación Veterinaria de Tandil (CIVETAN), Consejo Nacional de Investigaciones Científicas y Técnicas (CONICET), Tandil, Buenos Aires, Argentina

\* cceriani@vet.unicen.edu.ar

**Data Availability Statement:** All sequence files are available from the SRA database (accession number(s) SRR11713582, SRR11713583,

## Abstract

Bovine leukemia virus (BLV) is a δ-retrovirus responsible for Enzootic Bovine Leukosis (EBL), a lymphoproliferative disease that affects cattle. The virus causes immune system deregulation, favoring the development of secondary infections. In that context, mastitis incidence is believed to be increased in BLV infected cattle. The aim of this study was to analyze the transcriptome profile of a BLV infected mammary epithelial cell line (MAC-T). Our results show that BLV infected MAC-T cells have an altered expression of IFN I signal pathway and genes involved in defense response to virus, as well as a collagen catabolic process and some protooncogenes and tumor suppressor genes. Our results provide evidence to better understand the effect of BLV on bovine mammary epithelial cell's immune response.

## Introduction

Bovine leukemia virus (BLV) is an oncogenic virus that causes enzootic bovine leukemia in cattle. This virus is globally distributed, except for Europe, Australia, and New Zealand [1]. Most of the infected animals remain asymptomatic, while only 5–10% will develop lymphosarcomas, the final stage of the disease. The main target cell is the B lymphocyte IgM$^+$ CD5$^+$; nevertheless, it can also infect other cell types [2]. Due to its effects on immune cells, it has been associated with an increased incidence of secondary infections. As has been previously reported in the literature, BLV infection could cause a higher incidence of mastitis [3,4].

The mammary epithelium plays a critical role in the early defense against pathogens that causes mastitis. Besides being a physical barrier to pathogens, bovine mammary epithelial cells (bMEC) express pattern recognition receptors (PRRs), whose activation stimulates cytokine production and immune cell recruitment. Any disturbance in this biological system could impair the normal response of this epithelium to pathogen invasion. Recently, a bovine mammary epithelial cell line was stably infected with BLV *in vitro* (MAC-T BLV) [5], showing that

SRR11713584, SRR11713585, SRR11713586, SRR11713587.) after acceptance.

**Funding:** GLD. PICT 2016-4409 AGENCIA NACIONAL DE PROMOCION CIENTIFICA Y TECNOLOGICA. www.agencia.mincyt.gob.ar. MCC. PICT 2017-0378 AGENCIA NACIONAL DE PROMOCION CIENTIFICA Y TECNOLOGICA. www.agencia.mincyt.gob.ar. The funders had no role in study design, data collection and analysis, decision to publish, or preparation of the manuscript.

**Competing interests:** The authors have declared that no competing interests exist.

these cells are susceptible to productive BLV infection. Moreover, these cells showed reduced viability and TLR2 mRNA expression when exposed to heat-inactivated *S. aureus*, suggesting that the innate immune response could be impaired in BLV infected bMEC [6]. These findings signal the need for additional studies to better understand the effect of BLV infection on the bovine mammary epithelial immune response. The development of new transcriptome technologies allows us to analyze the global impact of BLV infection on bMEC. The objective was to analyze the transcription profile of BLV infected and uninfected bovine mammary epithelial cells (MAC-T) by next-generation sequencing (NGS). Our results show that BLV infection of the bovine mammary epithelial cell line MAC-T causes increased expression of genes related to the immune response, in addition to altered expression of genes associated with collagen catabolic process, protooncogenes and tumor suppressor genes. The information collected here could help researchers to better understand the viral effect on bMEC.

## Materials and methods

### Cell lines and culture conditions

MAC-T [7] (gift from Dr. Juan Loor from University of Illinois) and MAC-T stably infected with BLV (MAC-T BLV) cells were cultured in Modified Eagle's medium (MEM) supplemented with 10% fetal bovine serum (MIDSCI, Valley Park, MO, USA) and 1 μg/ml hydrocortisone (Sigma-Aldrich, Saint Louis, MO, USA) at 37˚C with 5% $CO_2$. MAC-T BLV cells were infected with PBMCs from BLV positive animals from Kansas, United States, as previously described [5]. Cultures were passaged upon approaching confluence using standard techniques, and MAC-T cells were always passed in the first place. MAC-T BLV cells were subcultured after MAC-T to avoid cross-contamination. For RNA seq analysis, MAC-T samples were obtained in passages 8, 30, and 36, while the three samples from MAC-T BLV were obtained in passage 35 post-infection (from 3 different infections with BLV). Each biological replicate consisted of MAC-T cells infected with the same batch of PBMCs obtained from a cow infected with BLV. All the cell lines were harvested for RNA extraction when they reached confluency.

### RNA extraction and sequencing

MAC-T and MAC-T BLV cells were grown in a six-well plate until a 100% confluency. After three washes with PBS, the cells were harvested with 600 μL of Trizol® (Thermo Fisher Scientific, Waltham, MA, USA). Total RNA was extracted from 3 samples of MAC-T and 3 samples of MAC-T BLV using the Direct-zol™ RNA MiniPrep Plus kit (Zymo Research, USA). The RNA quality was analyzed by Bioanalyzer (Agilent, USA), following the manufacturer´s instructions. Samples were sequenced by Novogene Corporation (https://en.novogene.com), who constructed the cDNA library using RNA-NEBNext Ultra RNA Library Prep Kit for Illumina (New England Biolabs, Ipswich, MA, USA). The sequencing performed in the NovaSeq 6000 (Illumina, San Diego, CA, USA) resulted in 150 pb paired-end sequences.

The data sent by Novogene Corporation were analyzed by our laboratory. FastQC analysis in Galaxy [8] showed that all the sequences had a score higher than 30, and it was not necessary to filter or trim them. The sequences were aligned against the bovine reference genome bosTau8 using the program HISAT2 [9]. The aligned counts were analyzed using GenomicAlignments and GenomicFeatures in R [10]. Differentially expressed genes were identified with DESeq2 in R [11]. The functional enrichment of differentially expressed genes was performed in Database for Annotation, Visualization, and Integrated Discovery (DAVID, versión 6.7; http://david.abcc.ncifcrf.gov). The principal component analysis, heat map, and volcano plot were done in the platform iDEP. Later, PCA was done in R using the gplots to allow label identification of the samples [12].

**Table 1. Summary of the sequences and alignments with the bovine reference genome BosTau8.**

| Sample | Total reads | Unique alignment | Unique alignment (%) | Multiple alignments | Multiple alignments (%) | No alignment | No alignment (%) |
|---|---|---|---|---|---|---|---|
| MAC-T 1 | 26023368 | 20893092 | 80.29 | 1853329 | 7.12 | 3276947 | 12.59 |
| MAC-T 2 | 25330865 | 20373892 | 80.43 | 1510347 | 5.96 | 3446626 | 13.61 |
| MAC-T 3 | 37707749 | 30657132 | 81.30 | 2014877 | 5.34 | 5035740 | 13.35 |
| MAC-T BLV 1 | 29518689 | 22753241 | 77.08 | 1935843 | 6.56 | 4829605 | 16.36 |
| MAC-T BLV 2 | 27188819 | 23570428 | 86.69 | 862148 | 3.17 | 2756243 | 10.14 |
| MAC-T BLV 3 | 29272389 | 25223952 | 86.17 | 919302 | 3.14 | 3129135 | 10.69 |

## Results

Comparative transcriptome analysis was performed in uninfected and BLV-infected bovine mammary epithelial cells (MAC-T). A total of 175.041.879 reads were obtained, with an average of 29.173.647 reads in each sample. Approximately 80% of MAC-T reads, and 83% of MAC-T BLV aligned uniquely with the reference genome (Table 1).

From a total of 14193 genes, only 9852 overcome the low expression filter (more than 0.5 CPM in at least one sample). The hierarchical clustering analysis done by the platform IDEP revealed that MAC-T samples segregate differently than MAC-T BLV, suggesting that BLV infection affects the bovine mammary epithelial cell´s transcriptome (Fig 1). Interestingly, one of the MAC-T BLV biological replicates differ from the other two samples, and this is probably a consequence of biological variations. Some differences are also present among MAC-T samples, probably because they came from a different number of culture passages.

Principal component analysis (PCA) is a linear transformation of the data showing the highest variability between the samples (Fig 2). The first component (PC1) that separates MAC-T from MAC-T BLV represents 40% of the genetic variation between samples and is related to ribosomal complex development and ribosomal RNA metabolism and process. As observed in the heat map, mact 1 sample is quite different from mact2 and mact 3; however, there is no difference in RNA quality between the samples (available in S1 Text).

DESeq2 was used to analyze differential gene expression. Genes were filtered by a minimum log2-fold change $\geq 2$ (log2FC), 0.05 adjusted p-value, and false discovery rate (FDR) lower than 0.05. With those parameters, 352 differentially expressed genes were found: 211 upregulated and 141 downregulated. Fig 3 displays a volcano plot representing the distribution of the differentially expressed genes. The list of the most differentially expressed genes is shown in S1 Table. The five most upregulated genes in MAC-T BLV were OAS1Z, VRTN, B4GALT5, SGCE y CTSL, while the most downregulated were ADAMTS2, TYRP1, GJB6, PMP22, TAGLN y MMP13.

DAVID was used to understand the biological functions of the differentially expressed genes (Fig 4). The Gene Ontology (GO) analysis indicated that the differentially expressed genes were involved in defense response to virus, negative regulation of viral genome replication, organelle fission, response to type I interferon, innate immune response, collagen catabolic process, cellular response to fibroblast growth factor stimulus, induction of positive chemotaxis, skin development and embryo implantation.

## Discussion and conclusion

Although some studies analyzed the differential gene expression in cell lines transfected with the viral gene *tax* and in BLV infected and uninfected blood cells using microarrays [13,14], several questions regarding the effect of BLV infection on the cell's transcriptome remain to be

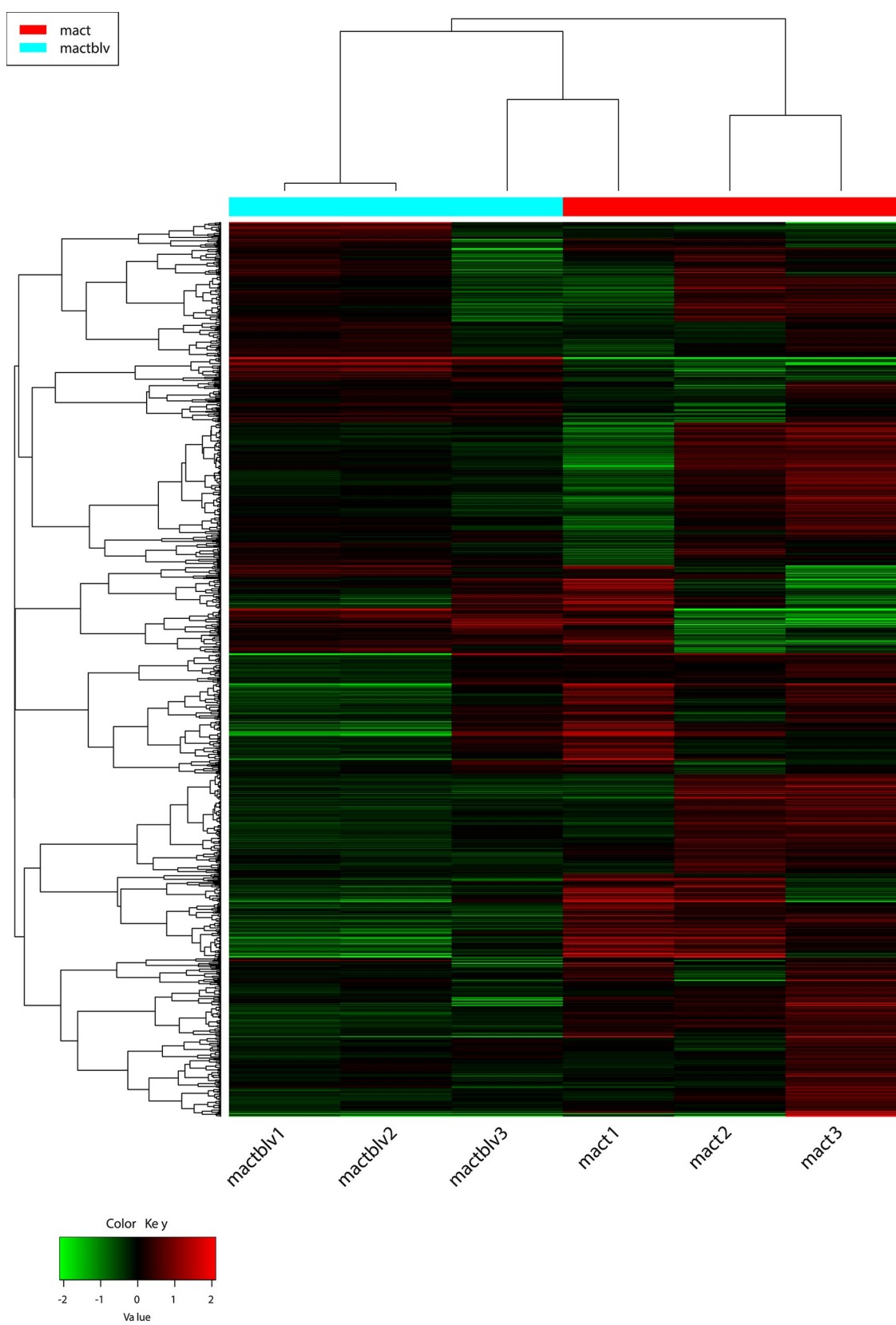

**Fig 1. Heatmap and hierarchical clustering based on the most differentially expressed genes.** The 1000 most differentially expressed genes between MAC-T (n = 3) and MAC-T BLV (n = 3) are shown.

addressed. To our knowledge, no prior studies have analyzed the effect of BLV infection on the cell´s transcriptome using RNA seq technology.

Particularly, the bMEC play a crucial role in the onset of the mammary gland immune response. We had previously infected these cells with BLV in order to analyze the effect of the virus on the response against pathogens that could cause mastitis. This is the first report of the transcriptome from bovine mammary epithelial cells (MAC-T) stably infected with BLV. A total of 352 genes were differentially expressed (log2FC $\geq$ 2, adj p-value $\leq$ 0.05, and FDR $\leq$ 0.05). Most of the upregulated genes were related to the host immune response, especially with IFN I signal. Type I IFN plays a crucial role in the antiviral immune response by

**Fig 2. PCA from MAC-T and MAC-T BLV samples.** Scatter plots of the first two principal components of the normalized gene expression profiles of MAC-T and MAC-T BLV. The PCA plot shows the variance of the three biological replicates of MAC-T and MAC-T BLV. Each axis represents the percentages of variation explained by the principal components.

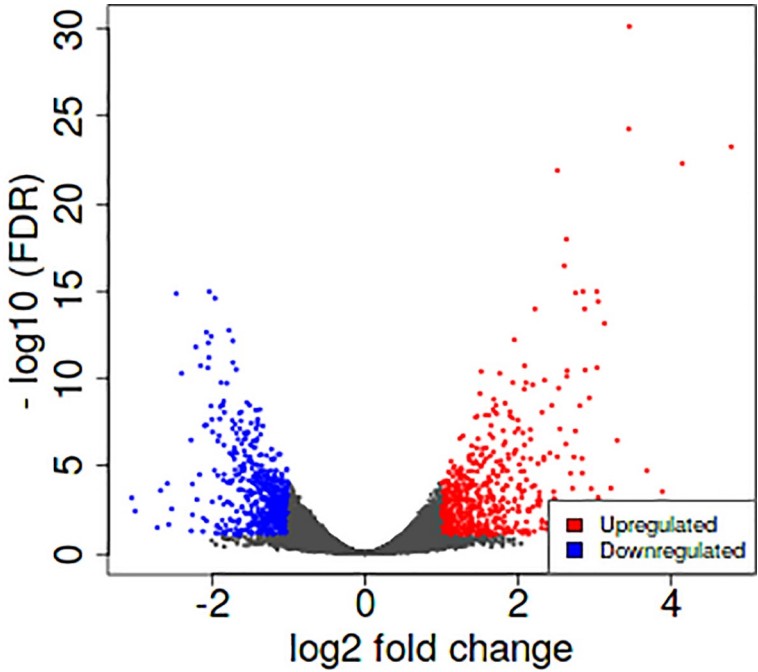

**Fig 3. Volcano plot representation of differentially expressed genes between MAC-T and MAC-T BLV.** Mean log2 fold change is plotted against the–log10 FDR for all the expressed genes. Red and blue dots represent genes with a significant increase or decrease expression, respectively.

stimulating the expression of the interferon-stimulated genes (ISG). Many ISG were upregulated in MAC-T BLV cells, including MX2, RSAD2, ISG15, OASL, IFI6, and IFI27. Also some chemokines such as CCL5 and CXCL8 were upregulated. On the other hand, genes related to the collagen catabolic process (SPARC, MMP13, and ADAMTS2) were downregulated.

ISG induce an antiviral response, which aims to block the viral replication in the different stages of the infection. These genes are also upregulated in other viral infections, including West Nile [15], Influenza A [16], and HIV [17]. It has been reported that the upregulation of these genes could cause an immunomodulatory effect that alters the capacity of the neutrophils to recognize and phagocyte pathogenic bacteria [18]. Moreover, overexpression of these ISG genes could be detrimental and increase the pathogenicity and susceptibility to bacterial infections [19–21]. These results suggest that bovine epithelial mammary cells persistently infected with BLV could have altered their capacity to react to other pathogens.

Epithelial cells are the first barrier in the immune response and express cytokines and chemokines that recruit immune cells to the site of infection. In MAC-T BLV cells, an increase in CCL5 and CXCL8 was observed. Overexpression of these chemokines highlights the important role of the bovine mammary epithelium in the onset of the antiviral immune response and has also been described in other viral infections [22–24].

One of the GO terms altered in MAC-T BLV cells is the collagen catabolic process. In this biological pathway, MAC-T BLV cells show a downregulation of genes, including SPARC, MMP19, and ADAMTS2. SPARC plays a role in collagen binding to the basal membrane, and it has been reported that a decrease in this protein is associated with tumor progression and metastasis in bladder cancer and leukemias [25,26]. MMP13 is a matrix metallopeptidase that cleaves type II collagen and participates in matrix remodeling from physiological and pathological processes [27]. ADAMST2 also plays a role in extracellular matrix remodeling. Its main

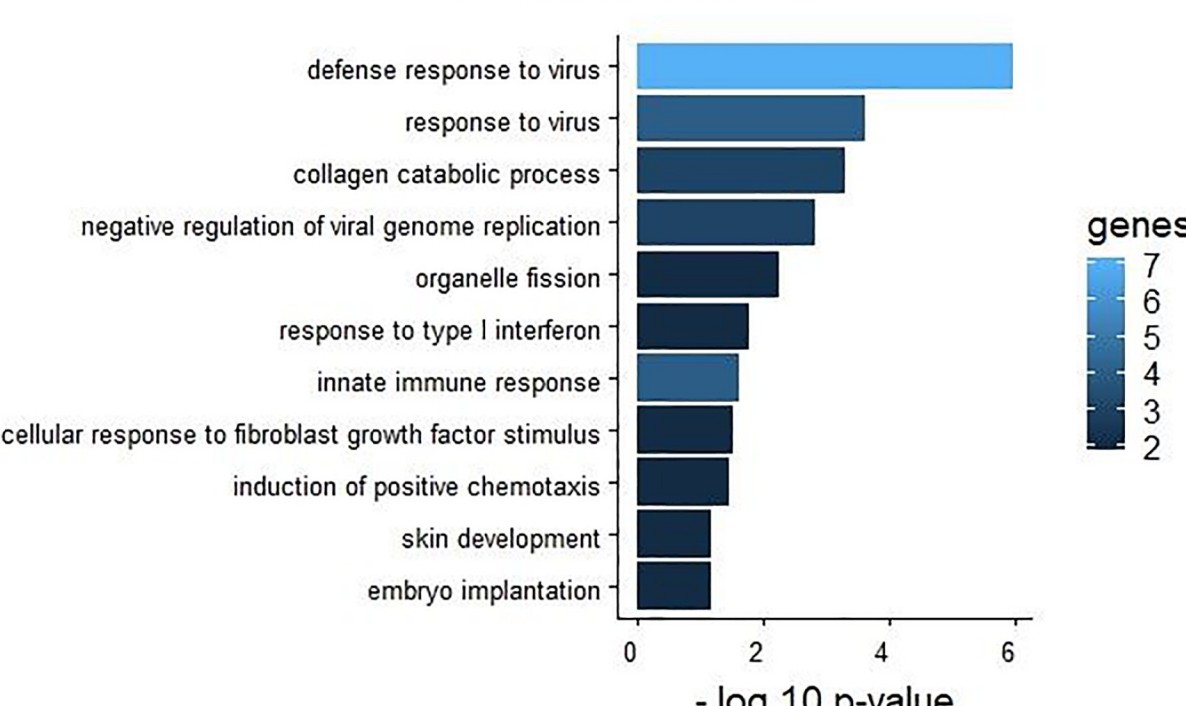

**Fig 4. Functional enrichment of differentially expressed genes.** Most significant GO terms associated with down and upregulated genes. The bar length indicates the significance and equals to the negative logarithm of the p-value. The bar color indicates the number of genes of that pathway that are differentially expressed in MAC-T BLV cells.

function is to clive collagen precursors, although it also has antitumoral and antiangiogenic properties [28]. The downregulation of these genes in MAC-T BLV could be involved in tumor development *in vivo*.

Among the differentially expressed genes in MAC-T BLV, some were related to cancer development, including ADGR1, NmU, PADI2, ID1, DPPA4 y XAF1. Considering that BLV is an oncogenic virus that causes lymphosarcomas, a deeper study of these genes could be useful for a better understanding of the virus pathogenicity and to develop new therapeutic strategies. It is well known that ADGRD1 is a G coupled receptor that plays a role in immune response, angiogenesis, and development [29]. Nmu is a neuropeptide that activates an inflammatory state by favoring cytokine expression and eosinophils activation. Overexpression of this protein has been associated with a worst outcome in cancer patients. Nmu overexpression induces cell proliferation, migration, invasion, and apoptosis resistance [30]. Moreover, PADI2 is an enzyme whose overexpression was associated with inflammatory process and tumor development [31]. ID1 is a DNA binding protein that has an active role in the control of cell proliferation and growth. Its increase has been reported in different types of cancer [32]. Many cell lines and cancer tissues present an increased expression of DPPA4, a gene that codifies for a nuclear factor essential during embryogenesis for pluripotential maintenance [33].

On the other hand, MAC-T BLV cells show a decreased expression of tumor suppressor genes, including transgrelin (TAGLN) and GJB6. TAGLN is an actin binding protein that is involved in cellular growth, migration, and matrix remodeling. It is believed to inhibit MMP9 activity, an enzyme necessary for tumor migration and invasion. TAGLN downregulation has

also been reported in breast cancer [34]. Moreover, GJB6 participates in cell cycle regulation and acts as a tumor suppressor gene. Its decreased expression is associated with tumor progression [35]. The downregulation of these genes could play a role in tumor development *in vivo* in BLV infected animals. These findings signal the need for additional studies to understand more about the genes implicated in tumor development in BLV infected animals.

Although we found a differential expression of some oncogenes and tumor suppressor genes in this study, there are almost no reports of breast cancer in cattle. BLV infection has been mainly studied in dairy cattle where the animals have high rates of pregnancies that are believed to be protective against breast cancer [36]. Notwithstanding, many studies associate BLV with human breast cancer development [36]. It would be interesting to analyze if the genes found in this study are also differentially expressed in samples from human breast cancer where BLV fragments were detected. Moreover, it is well known that chronic inflammation is associated with cancer development [37]. In that context, the study of the immune-related genes in which altered expression was observed in MAC-T BLV cells would also be interesting.

The PCA analysis showed that 40% of the genetic variation between samples is related to ribosomal complex development and ribosomal RNA metabolism and process, which is active in multiple neoplastic cells with highly increased protein synthesis.[38] Ribosomal complex development and ribosomal RNA metabolism and process are component of the ribosome biogenesis, a process needed for cell growth and proliferation. It is well known that people with defects in ribosome biogenesis (called ribosomopathies) are at an increased risk of developing cancer. Moreover, there are some linkages between the neoplastic transformation of chronically inflamed tissues and alterations in ribosome biogenesis [39]. Since BLV could be a cause of chronic inflammation, this new information about the alteration in ribosome biogenesis opens a new perspective to analyze the neoplastic effect of BLV infection. Further studies are needed to understand how these factors could contribute to BLV tumor development.

A previous study using microarrays analyzed the differential gene expression in a HeLa cell line transfected with the BLV Tax protein[13]. Interestingly and contrary to our findings, they reported that Tax expressing cells decreased the expression of genes involved in the immune response. Particularly, among the downregulated immune genes many were related to the interferon family of anti-viral genes, such as ISG15 and OASL, which we found to be upregulated. The reason for these contradictory results could be the many differences in the studies designs. For instance, in the study performed by Arainga and collaborators they used a human cell line that is not the natural host of the virus; moreover, they only studied the effect of the BLV transactivating protein Tax, while our study analyze the effect of the virus infection in bovine epithelial mammary cells. What is more, in Arainga´s study, differential gene expression was analyzed 24 h after the cells were transfected with Tax, while in our study, RNA extraction was performed on cells at 35 days post-infection with BLV. This is an important difference since we are analyzing the effect of the virus in the long term. BLV is known to cause persistent infections and its main clinical manifestation (that are the lymphosarcomas) appear from 5 to 10 years after the infection. Our intention was to measure the effect of the virus on the cell lines persistently exposed to the virus to see how it could affect the immune function of those cells and to evaluate if there were differentially expressed genes that could play a role in the neoplastic process that leads to the lymphosarcomas.

A microarrays study performed in 2017 by Brym and Kamiński on whole blood from BLV infected and uninfected cattle shows that the virus induced differential expression of genes related to the innate immune response and the neoplastic transformation [14]. Although we found some similar enriched pathways in our study, most of our differentially expressed genes were upregulated, in contrast with Brym´s results, where a greater number of downregulated genes were found. In line with that disparity, the only two differentially expressed genes

present in both studies had different expressions: CXCL8 and CTSL were downregulated in Brym's study and upregulated in ours. It is believed that BLV remains silent in PBMCs *in vivo*, and *ex vivo* incubation of those cells immediately activates the virus expression. In Brym's study, samples were inactivated right after bleeding, which means that the results they found probably reflect what is happening *in vivo* in BLV silently infected cells. On the other hand, our study was performed in persistently infected BLV cell lines that actively release the virus. The amount of viral sequences identified in each MAC-T BLV sample is available in S2 Table. Our study probably reflects the stage of the disease where there is an increased viral expression. Although many neoplastic related genes were found, none of them were shared between both studies.

In summary, this *in vitro* transcriptome analysis gives further insight into the BLV infection effect on a bovine mammary epithelial cell line, highlighting important genes in the viral immune response and the tumorigenic process. The genes identified in this study could be potential targets of intervention for vaccine development or therapeutic strategies. Moreover, a further study of those genes could help to elucidate if BLV is related to human breast cancer. On the other hand, our findings demonstrate an upregulation of many interferon-induced genes that could be affecting the mammary epithelial immune response. Previously, the increase of IFN I pathway was associated with greater susceptibility to bacterial infection. Taking that into account, future research should investigate how the increase of IFN I pathway on bMEC could affect the local immune response against pathogens causative of mastitis.

## Supporting information

**S1 Text. Samples RNA quality.**
(PDF)

**S1 Table. Most differentially expressed genes between MAC-T and MAC-T BLV.**
(DOCX)

**S2 Table. Summary of the sequences aligned with BLV reference genome AP018032.**
(DOCX)

## Author Contributions

**Conceptualization:** Lucia Martinez Cuesta, Guillermina Laura Dolcini, Maria Carolina Ceriani.

**Formal analysis:** Lucia Martinez Cuesta, Juan Pedro Liron, Guillermina Laura Dolcini, Maria Carolina Ceriani.

**Funding acquisition:** Guillermina Laura Dolcini, Maria Carolina Ceriani.

**Investigation:** Lucia Martinez Cuesta, María Victoria Nieto Farias, Maria Carolina Ceriani.

**Methodology:** Lucia Martinez Cuesta, Guillermina Laura Dolcini.

**Project administration:** Maria Carolina Ceriani.

**Resources:** Guillermina Laura Dolcini, Maria Carolina Ceriani.

**Software:** Juan Pedro Liron.

**Supervision:** Maria Carolina Ceriani.

**Validation:** Lucia Martinez Cuesta, Juan Pedro Liron.

**Visualization:** Lucia Martinez Cuesta.

**Writing – original draft:** Lucia Martinez Cuesta.

**Writing – review & editing:** Lucia Martinez Cuesta, Juan Pedro Liron, Guillermina Laura Dolcini, Maria Carolina Ceriani.

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
