## [Decision Letter · Decision Letter 0]

30 Apr 2020

PONE-D-20-07350

“Effect of bovine leukemia virus (BLV) infection on bovine mammary epithelial cells transcriptome profile.”

PLOS ONE

Dear Dr. Ceriani,

Thank you for submitting your manuscript to PLOS ONE. After careful consideration, we feel that it has merit but does not fully meet PLOS ONE’s publication criteria as it currently stands. Therefore, we invite you to submit a revised version of the manuscript that addresses the points raised during the review process.

We would appreciate receiving your revised manuscript by Jun 14 2020 11:59PM. To enhance the reproducibility of your results, we recommend that if applicable you deposit your laboratory protocols in protocols.io, where a protocol can be assigned its own identifier (DOI) such that it can be cited independently in the future. For instructions see: http://journals.plos.org/plosone/s/submission-guidelines#loc-laboratory-protocols

We look forward to receiving your revised manuscript.

Kind regards,

Maria del Mar Ortega-Villaizan

Academic Editor

PLOS ONE

Additional Editor Comments (if provided):

The manuscript should be deeply revised before being accepted for publicaction. Major revisions are needed.

2. In your Methods section, please provide additional details regarding each of the cell lines used in your study, including any quality control testing procedures, and ensure you have described the source. For more information regarding PLOS' policy on materials sharing and reporting, see https://journals.plos.org/plosone/s/materials-and-software-sharing#loc-sharing-materials, and for more information on PLOS ONE's guidelines for research using cell lines, see https://journals.plos.org/plosone/s/submission-guidelines#loc-cell-lines.

3. We note that you are reporting an analysis of a microarray, next-generation sequencing, or deep sequencing data set. PLOS requires that authors comply with field-specific standards for preparation, recording, and deposition of data in repositories appropriate to their field. Please upload these data to a stable, public repository (such as ArrayExpress, Gene Expression Omnibus (GEO), DNA Data Bank of Japan (DDBJ), NCBI GenBank, NCBI Sequence Read Archive, or EMBL Nucleotide Sequence Database (ENA)). In your revised cover letter, please provide the relevant accession numbers that may be used to access these data. For a full list of recommended repositories, see http://journals.plos.org/plosone/s/data-availability#loc-omics or http://journals.plos.org/plosone/s/data-availability#loc-sequencing.

Reviewers' comments:

Reviewer's Responses to Questions

**Comments to the Author**

1. Is the manuscript technically sound, and do the data support the conclusions?

Reviewer #1: Partly

Reviewer #2: Partly

2. Has the statistical analysis been performed appropriately and rigorously? 

Reviewer #1: N/A

Reviewer #2: I Don't Know

3. Have the authors made all data underlying the findings in their manuscript fully available?

Reviewer #1: Yes

Reviewer #2: Yes

4. Is the manuscript presented in an intelligible fashion and written in standard English?

Reviewer #1: Yes

Reviewer #2: Yes

5. Review Comments to the Author

Reviewer #1: This group published two BLV-related manuscript in 2018 (Stable infection of a bovine mammary epithelial cell line (MAC-T) with bovine leukemia virus (BLV).), and in 2019 (Effect of bovine leukemia virus on bovine mammary epithelial cells.). This work is a continuation of the previous studies while RNAseq was performed to observe the impact of BLV infection on the transcriptome of bovine mammary epithelial cells.

The authors are encouraged to revise this manuscript so that the readers can understand and follow your manuscript well. Each figure should have its own detailed figure legend. You do not have to leave the Figures in the end of this manuscript while put the title of figure inside of the main text. Large table such as Table-2 can be moved to the supplementary Tables.

This manuscript has two parts: the part-1 is about the RNAseq, and the part-2 is related to the BLV sequencing. It is my opinion that the authors may consider to remove the part-2 totally from this manuscript. You obtained a viral sequence of 8419 pb that represents 96.6% of the complete genome, and this is not enough for BLV as a retrovirus. Then, you use the sequence with 96.8% of coverage to perform phylogenetic comparison, and it is not valid to have the statement (showed a 99% homology with reported Japanese viral genomes).

Please focus on the part-1 of your research, and come up with more specific take-home messages from the analysis of the RNAseq (more information that what you have now: BLV infected MAC-T cells have an altered expression of INF I signal pathway and genes involved in defense response to virus, as well as a collagen catabolic process and some protooncogenes and tumor suppressor genes).

Reviewer #2: Title- just suggestion - to add that the transcriptomic analysis was done with the RNAseq method would be more informative

Abstract - line 21 - should be: the RNA seq data

Materials and methods

- Preparation of experimental material should be more precisely written- how the biological replicate of the experiment were prepared? Were the cultures of MAC-T and MAC-T BLV conducted and harvested in parallel? Was total RNA isolated from each repetition at the same time? These questions will be reflected in the comments on the results.

- The RNA seq data were prepared by the sequencing platform - where? any company name? If whole bioinformatic analysis were done by Authors or any Center for Informatics?

As far as I know for PLOS ONE Authors contributions is required

Results

Fig.1 and line 87-89 - will be more informative which program was used for compute the heat map ans some datails for scale bar is missing.

It is true that hierarchical analysis revealed that MAC-T samples segregate differently than MAC-T BLV but also showes that intra-group variability is high, only mactblv1 and mactblv2 are similar. In the results section RIN data are missing. Here the way of preparation of experimental material could be significant. The number of 352 genes differentially expressed might be real but there is some kind of doubt that intra-group variability can affect this. Please comment and explain...

Fig.2 and line 92-95

Will be more informative when samples on the PCA plot have the same, corresponding name as in the heat map as mactblv1....It seems that the top one red point probably showed differential RNA-seq quality? Please comment...

There is an information that the genetic variation between samples is related to ribosomal complex development

and ribosomal RNA metabolism and process- commentary on these results is not visible in the discussion. Why?

Line 97-98, tab.2 and 138 -the data should be compatible fold change or log2FC?

Line 123-126 and Figure 5 - Is not the aim of the study but if BLV sequence obtained from MAC-T BLV has been already phylogenetically characterised, it would be interesting to classify it in terms of genotypes. Moreover, the tree could also contain sequences from the United States. On a tree there is a mistake - the sequence LC164083 is not a Sheep but come from FLK-BLV cells.

Discussion and conclussion

Line 145 West Nile

It is surprising that Authors do not discuss their results with thealready mentioned papers Arainga et al and Brym, Kaminski; for example the CXCL8 proteins was also mentioned as DEG at Brym paper. The authors themselves have already done some expressive RTqPCR tests on this line - whether there was any reflection of the results in RNAseq? - for example TLR9.

6. PLOS authors have the option to publish the peer review history of their article (what does this mean?). If published, this will include your full peer review and any attached files.

Reviewer #1: Yes: Chengming Wang

Reviewer #2: No

---

## [Author Response · Author response to Decision Letter 0]

7 May 2020

Reviewer #1: 

This group published two BLV-related manuscript in 2018 (Stable infection of a bovine mammary epithelial cell line (MAC-T) with bovine leukemia virus (BLV).), and in 2019 (Effect of bovine leukemia virus on bovine mammary epithelial cells.). This work is a continuation of the previous studies while RNAseq was performed to observe the impact of BLV infection on the transcriptome of bovine mammary epithelial cells.

The authors are encouraged to revise this manuscript so that the readers can understand and follow your manuscript well. Each figure should have its own detailed figure legend. You do not have to leave the Figures in the end of this manuscript while put the title of figure inside of the main text. Large table such as Table-2 can be moved to the supplementary Tables.

Following your suggestion Table 2 was moved to supplementary information. All figures are now submitted in an individual file as requested by the journal and the captions in the manuscript have a detailed legend.

This manuscript has two parts: the part-1 is about the RNAseq, and the part-2 is related to the BLV sequencing. It is my opinion that the authors may consider to remove the part-2 totally from this manuscript. You obtained a viral sequence of 8419 pb that represents 96.6% of the complete genome, and this is not enough for BLV as a retrovirus. Then, you use the sequence with 96.8% of coverage to perform phylogenetic comparison, and it is not valid to have the statement (showed a 99% homology with reported Japanese viral genomes).

Following your suggestion the BLV sequencing was removed from the manuscript.

Please focus on the part-1 of your research, and come up with more specific take-home messages from the analysis of the RNAseq (more information that what you have now: BLV infected MAC-T cells have an altered expression of INF I signal pathway and genes involved in defense response to virus, as well as a collagen catabolic process and some protooncogenes and tumor suppressor genes).

Thank you for this suggestion. We enriched the discussion section and specially the last summarizing paragraph in order to give a better take home message.

Reviewer #2: 

Title- just suggestion - to add that the transcriptomic analysis was done with the RNAseq method would be more informative 

Taking your suggestion, the title was changed to “Effect of bovine leukemia virus (BLV) infection on bovine mammary epithelial cells RNA-seq transcriptome profile.”

Abstract - line 21 - should be: the RNA seq data. 

Thanks for the correction. All the information regarding the genome sequence obtained by RNA seq was deleted following reviewer 1 suggestion.

Materials and methods

- Preparation of experimental material should be more precisely written- how the biological replicate of the experiment were prepared? Were the cultures of MAC-T and MAC-T BLV conducted and harvested in parallel? Was total RNA isolated from each repetition at the same time? These questions will be reflected in the comments on the results. 

This are very good observations. We increased the details on the sample preparation answering your specific questions. how the biological replicate of the experiment were prepared? Each biological replicate consisted of MAC-T cells infected with the same batch of PBMCs obtained from a cow infected with BLV. Were the cultures of MAC-T and MAC-T BLV conducted and harvested in parallel? Cultures were passaged upon approaching confluence using standard techniques and MAC-T cells were always passed in first place. MAC-T BLV cells were subcultured after MAC-T to avoid cross-contamination. Was total RNA isolated from each repetition at the same time? No. All the cell lines were harvested for RNA extraction when they reached confluency.

- The RNA seq data were prepared by the sequencing platform - where? any company name? If whole bioinformatic analysis were done by Authors or any Center for Informatics? 

Thank you for highlighting this, we recognize it was not clear in the original manuscript. I copy the new statement: “Samples were sequenced by Novogene Corporation (https://en.novogene.com), who constructed the cDNA library by using RNA - NEBNext Ultra RNA Library Prep Kit for Illumina (New England Biolabs, Ipswich, MA, USA). The sequencing performed in the NovaSeq 6000 (Illumina, San Diego, CA, USA) resulted in 150 pb paired-end sequences. The analysis of the sequencing images and the Fasta file creation was done with the software Casava v1.8.2 (Illumina, San Diego, CA, USA). The data sent by Novogene Corporation were analyzed by our laboratory.”

As far as I know for PLOS ONE Authors contributions is required. 

A file with each author`s contribution is attached with the manuscript.

Results

Fig.1 and line 87-89 - will be more informative which program was used for compute the heat map ans some datails for scale bar is missing. 

The program used for the analysis is now indicated clearly in the following sentence: “The hierarchical clustering analysis done by the platform IDEP revealed that MAC-T samples segregate differently than MAC-T BLV, suggesting that BLV infection affects the bovine mammary epithelial cell´s transcriptome (Fig 1).” And a new picture was obtained where the scale bar is more readable. 

It is true that hierarchical analysis revealed that MAC-T samples segregate differently than MAC-T BLV but also showes that intra-group variability is high, only mactblv1 and mactblv2 are similar. In the results section RIN data are missing. Here the way of preparation of experimental material could be significant. The number of 352 genes differentially expressed might be real but there is some kind of doubt that intra-group variability can affect this. Please comment and explain... 

The RIN data is now added as supplementary material. The RIN is not significantly different among samples from MAC-T or MAC-T BLV groups. The difference could be due to biological variations. Particularly, in MAC-T group the difference observed could be a cause of the different passages used for the analysis.

Fig.2 and line 92-95

Will be more informative when samples on the PCA plot have the same, corresponding name as in the heat map as mactblv1....It seems that the top one red point probably showed differential RNA-seq quality? Please comment...

The figure is modified and now shows the names of the samples. We added a comment in the manuscript that states: “ As observed in the heat map, mact 1 sample is quite different from mact2 and mact 3; however, there is no difference in RNA quality between the samples (available in S1 Text).”

There is an information that the genetic variation between samples is related to ribosomal complex development

and ribosomal RNA metabolism and process- commentary on these results is not visible in the discussion. Why?

We are sorry we did not include a comment on that finding. We now added a paragraph in the discussion section about that: “The PCA analysis showed that 40% of the genetic variation between samples is related to ribosomal complex development and ribosomal RNA metabolism and process, which is active in multiple neoplastic cells with highly increased protein synthesis.(37) Ribosomal complex development and ribosomal RNA metabolism and process are component of the ribosome biogenesis, a process needed for cell growth and proliferation. It is well known that people with defects in ribosome biogenesis (called ribosomopathies) are at an increased risk of developing cancer. Moreover, there are some linkages between neoplastic transformation of chronically inflammated tissues and alterations in ribosome biogenesis (38). Since BLV could be a cause of chronic inflammation, this new information about the alteration in ribosome biogenesis opens a new perspective to analyze the neoplastic effect of BLV infection. Further studies are needed to understand how these factors could contribute to BLV tumor development.”

Line 97-98, tab.2 and 138 -the data should be compatible fold change or log2FC?

It is log2FC in all cases. We corrected that in materials and methods.

Line 123-126 and Figure 5 - Is not the aim of the study but if BLV sequence obtained from MAC-T BLV has been already phylogenetically characterised, it would be interesting to classify it in terms of genotypes. Moreover, the tree could also contain sequences from the United States. On a tree there is a mistake - the sequence LC164083 is not a Sheep but come from FLK-BLV cells.

The phylogenetic analysis was removed from the manuscript following Reviewer 1 suggestion.

Discussion and conclussion

Line 145 West Nile. Modified 

It is surprising that Authors do not discuss their results with thealready mentioned papers Arainga et al and Brym, Kaminski; for example the CXCL8 proteins was also mentioned as DEG at Brym paper. The authors themselves have already done some expressive RTqPCR tests on this line - whether there was any reflection of the results in RNAseq? - for example TLR9.

Thank you for this suggestion. We specifically added two paragraphs to discuss the results by Arainga and Brym. However, we excluded a discussion about our previous results because those were done in a different infectious stage.

---

## [Decision Letter · Decision Letter 1]

5 Jun 2020

“Effect of bovine leukemia virus (BLV) infection on bovine mammary epithelial cells RNA-seq transcriptome profile.”

PONE-D-20-07350R1

Dear Dr. Ceriani,

We’re pleased to inform you that your manuscript has been judged scientifically suitable for publication and will be formally accepted for publication once it meets all outstanding technical requirements.

Kind regards,

Maria del Mar Ortega-Villaizan

Academic Editor

PLOS ONE

Additional Editor Comments (optional):

The reviewers comments have been adressed. Just check one more correction by Reviewer 2. The manuscript is ready for publication.

Reviewers' comments:

Reviewer's Responses to Questions

**Comments to the Author**

1. If the authors have adequately addressed your comments raised in a previous round of review and you feel that this manuscript is now acceptable for publication, you may indicate that here to bypass the “Comments to the Author” section, enter your conflict of interest statement in the “Confidential to Editor” section, and submit your "Accept" recommendation.

Reviewer #1: All comments have been addressed

Reviewer #2: All comments have been addressed

2. Is the manuscript technically sound, and do the data support the conclusions?

Reviewer #1: Yes

Reviewer #2: Yes

3. Has the statistical analysis been performed appropriately and rigorously? 

Reviewer #1: N/A

Reviewer #2: Yes

4. Have the authors made all data underlying the findings in their manuscript fully available?

Reviewer #1: Yes

Reviewer #2: Yes

5. Is the manuscript presented in an intelligible fashion and written in standard English?

Reviewer #1: Yes

Reviewer #2: Yes

6. Review Comments to the Author

Reviewer #1: Appreciate the efforts from the authors to revise this manuscript. A great job by the authors to address this reviewer's comments.

Reviewer #2: Please check - line 59 - passage 35 and line 214 - 35 days after infection - whether this is compatible with each other?

7. PLOS authors have the option to publish the peer review history of their article (what does this mean?). If published, this will include your full peer review and any attached files.

Reviewer #1: Yes: Chengming Wang

Reviewer #2: No

---

## [Editor Report · Acceptance letter]

11 Jun 2020

PONE-D-20-07350R1 

“Effect of bovine leukemia virus (BLV) infection on bovine mammary epithelial cells RNA-seq transcriptome profile.” 

Dear Dr. Ceriani:

I'm pleased to inform you that your manuscript has been deemed suitable for publication in PLOS ONE. Congratulations! Your manuscript is now with our production department. 

Kind regards, 

on behalf of

Dr. Maria del Mar Ortega-Villaizan 

Academic Editor

PLOS ONE